# Uncertainty Quantification for Language Models: A Suite of Black-Box, White-Box, LLM Judge, and Ensemble Scorers

**Dylan Bouchard**                                           *dylan.bouchard@cvshealth.com*
*CVS Health, Wellesley, MA*

**Mohit Singh Chauhan**                              *mohitsingh.chauhan@cvshealth.com*
*CVS Health, Wellesley, MA*

**Reviewed on OpenReview:** *https://openreview.net/forum?id=WOFspd4lq5*

## Abstract

Hallucinations are a persistent problem with Large Language Models (LLMs). As these models become increasingly used in high-stakes domains, such as healthcare and finance, the need for effective hallucination detection is crucial. To this end, we outline a versatile framework for closed-book hallucination detection that practitioners can apply to real-world use cases. To achieve this, we adapt a variety of existing uncertainty quantification (UQ) techniques, including black-box UQ, white-box UQ, and LLM-as-a-Judge, transforming them as necessary into standardized response-level confidence scores ranging from 0 to 1. To enhance flexibility, we propose a tunable ensemble approach that incorporates any combination of the individual confidence scores. This approach enables practitioners to optimize the ensemble for a specific use case for improved performance. To streamline implementation, the full suite of scorers is offered in this paper's companion Python toolkit, `uqlm`. To evaluate the performance of the various scorers, we conduct an extensive set of experiments using several LLM question-answering benchmarks. We find that our tunable ensemble typically surpasses its individual components and outperforms existing hallucination detection methods. Our results demonstrate the benefits of customized hallucination detection strategies for improving the accuracy and reliability of LLMs.

## 1 Introduction

Large language models (LLMs) are being increasingly used in production-level applications, often in high-stakes domains such as healthcare or finance. Consequently, there is a rising need to monitor these systems for the accuracy and factual correctness of model outputs. In these sensitive use cases, even minor errors can pose serious safety risks while also leading to high financial costs and reputational damage. A particularly concerning risk for LLMs is hallucination, where LLM outputs sound plausible but contain content that is factually incorrect. Many studies have investigated hallucination risk for LLMs (see Huang et al. (2023); Tonmoy et al. (2024); Shorinwa et al. (2024); Huang et al. (2024) for surveys of the literature). Even recent models, such as OpenAI's GPT-4.5, have been found to hallucinate as often as 37.1% on certain benchmarks (OpenAI, 2025), underscoring the ongoing challenge of ensuring reliability in LLM outputs.

Hallucination detection methods typically involve comparing ground truth texts to generated content, comparing source content to generated content, or quantifying uncertainty. Assessments that compare ground truth texts to generated content are typically conducted pre-deployment in order to quantify hallucination risk of an LLM for a particular use case. While important, this collection of techniques does not lend itself well to real-time evaluation and monitoring of systems already deployed to production. In contrast, techniques that compare source content to generated content or quantify uncertainty can compute response-level scores at generation time and hence can be used for real-time monitoring of production-level applications.

Uncertainty quantification (UQ) techniques can be used for hallucination detection in a closed-book setting, meaning they do not require access to a database of source content, ground truth texts, or internet access. These approaches are typically classified as either black-box UQ, white-box UQ, or LLM-as-a-Judge. Black-box UQ methods exploit the stochastic nature of LLMs and measure semantic consistency of multiple responses generated from the same prompt. White-box UQ methods leverage token probabilities associated with the LLM outputs to compute uncertainty or confidence scores. LLM-as-a-Judge methods use one or more LLMs to evaluate the factual correctness of a question-answer concatenation.

In this paper, we outline a versatile framework for generation-time, closed-book hallucination detection that practitioners can apply to real-world use cases. To achieve this, we adapt a variety of existing black-box UQ, white-box UQ, and LLM-as-a-Judge methods, applying transformations as necessary to obtain standardized confidence scores that range from 0 to 1.[1] For improved customization, we propose a tunable ensemble approach that incorporates any combination of the individual scorers. The ensemble output is a simple weighted average of these individual components, where the weights can be tuned using a user-provided set of graded LLM responses. This approach enables practitioners to optimize the ensemble for a specific use case, leading to more accurate and reliable hallucination detection. Importantly, our ensemble is extensible, meaning practitioners can expand to include new components as research on hallucination detection evolves.

We evaluate the full suite of UQ scorers on responses generated by four LLMs across various question-answer benchmarks, yielding several empirical insights. Most notably, our tunable ensemble generally outperforms its individual components for hallucination detection. However, the performance ranking of individual scorers varies by dataset, underscoring the value of tailoring methods to specific use cases. Among black-box UQ scorers, entailment-style approaches demonstrate superior performance in our comparisons. Furthermore, gains from sampling additional responses shrink as the number of candidate responses rises, providing practical guidance for deployment. Lastly, a model's accuracy on a given dataset positively relates to its performance as a judge of other models' answers on that dataset.

Finally, this paper is complemented by our open-source Python package, `uqlm`, that provides ready-to-use implementations of all uncertainty quantification methods presented and evaluated in this work.[2] `uqlm` enables practitioners to generate responses and obtain response-level confidence scores by providing prompts (i.e., the questions or tasks for the LLM) along with their chosen LLM. Our framework and toolkit provide researchers and developers a model-agnostic, user-friendly way to implement our suite of UQ-based scorers in real-world use cases, enabling more informed decisions around LLM outputs.

## 2 Related Work

**Black-Box UQ**  Cole et al. (2023) propose evaluating similarity between an original response and candidate responses using exact match-based metrics. In particular, they propose two metrics: repetition, which measures the proportion of candidate responses that match the original response, and diversity, which penalizes a higher proportion of unique responses in the set of candidates. These metrics have the disadvantage of penalizing minor phrasing differences even if two responses have the same meaning. Text similarity metrics assess response consistency in a less stringent manner. Manakul et al. (2023) propose using n-gram-based evaluation to evaluate text similarity. Similar metrics such as ROUGE (Lin, 2004), BLEU (Papineni et al., 2002), and METEOR (Banerjee & Lavie, 2005) have also been proposed (Shorinwa et al., 2024). These metrics, while widely adopted, have the disadvantage of being highly sensitive to token sequence orderings and often fail to detect semantic equivalence when two texts have different phrasing. Sentence embedding-based metrics such as cosine similarity (Qurashi et al., 2020), computed using a sentence transformer such as Sentence-Bert (Reimers & Gurevych, 2019), have also been proposed (Shorinwa et al., 2024). These metrics have the advantage of being able to detect semantic similarity in a pair of texts that are phrased differently. In a similar vein, Manakul et al. (2023) propose using BERTScore (Zhang et al., 2020), based

---

[1]We employ UQ-style signals as a means to produce response-level confidence for hallucination detection, rather than pursuing uncertainty quantification as an end in itself. We do not separate aleatoric and epistemic uncertainty.

[2]The `uqlm` repository can be found at `https://github.com/cvs-health/uqlm`. For a detailed description of the software, we refer the reader to Bouchard et al. (2025).

on the maximum cosine similarity of contextualized word embeddings between token pairs in two candidate texts.

Natural Language Inference (NLI) models are another popular method for evaluating similarity between an original response and candidate responses. These models classify a pair of texts as either *entailment*, *contradiction*, or *neutral*. Several studies propose using NLI estimates of $1 - P(\text{contradiction})$ or $P(\text{entailment})$ between the original response and a set of candidate responses to quantify uncertainty (Chen & Mueller, 2023; Lin et al., 2024). Zhang et al. (2024) follow a similar approach but instead average across sentences and exclude $P(\text{neutral})$ from their calculations.[3] Other studies compute semantic entropy using NLI-based clustering (Kuhn et al., 2023; Kossen et al., 2024; Farquhar et al., 2024). Qiu & Miikkulainen (2024) estimate density in semantic space for candidate responses.

**White-Box UQ**   Manakul et al. (2023) consider two scores for quantifying uncertainty with token probabilities: average negative log probability and maximum negative log probability. While these approaches effectively represent a measure of uncertainty, they lack ease of interpretation, are unbounded, and are more useful for ranking than interpreting a standalone score. Fadeeva et al. (2024) consider perplexity, calculated as the exponential of average negative log probability. Similar to average negative log probability, perplexity also has the disadvantage of being unbounded. They also consider response improbability, computed as the complement of the joint token probability of all tokens in the response. Although this metric is bounded and easy to interpret, it penalizes longer token sequences relative to semantically equivalent, shorter token sequences. Another popular metric is entropy, which considers token probabilities over all possible token choices in a pre-defined vocabulary (Malinin & Gales, 2021; Manakul et al., 2023). Malinin & Gales (2021) also consider the geometric mean of token probabilities for a response, which has the advantage of being bounded and easy to interpret.[4]

**LLM-as-a-Judge**   For uncertainty quantification, several studies concatenate a question-answer pair and ask an LLM to score or classify the answer's correctness. Chen & Mueller (2023) propose using an LLM for self-reflection certainty, where the same LLM is used to judge correctness of the response. Specifically, the LLM is asked to score the response as incorrect, uncertain, or correct, which map to scores of 0, 0.5, and 1, respectively. Similarly, Kadavath et al. (2022) ask the same LLM to state $P(Correct)$ given a question-answer concatenation. Xiong et al. (2024) explore several variations of similar prompting strategies for LLM self-evaluation. More complex variations such as multiple choice question answering generation (Manakul et al., 2023), multi-LLM interaction (Cohen et al., 2023), and follow-up questions (Agrawal et al., 2024) have also been proposed.

**Ensemble Approaches**   Chen & Mueller (2023) propose a two-component ensemble for zero-resource hallucination known as BSDetector. The first component, known as observed consistency, computes a weighted average of two comparison scores between an original response and a set of candidate responses, one based on exact match, and another based on NLI-estimated contradiction probabilities. The second component is self-reflection certainty, which uses the same LLM to judge correctness of the response. In their ensemble, response-level confidence scores are computed using a weighted average of observed consistency and self-reflection certainty. Verga et al. (2024) propose using a Panel of LLM evaluators (PoLL) to assess LLM responses. Rather than using a single large LLM as a judge, their approach leverages a panel of smaller LLMs. Their experiments find that PoLL outperforms large LLM judges, having less intra-model bias in the judgments.

---

[3] Averaging across sentences is done to address long-form responses. Jiang et al. (2024) also address long-form hallucination detection but follow a graph-based approach instead.

[4] For additional white-box uncertainty quantification techniques, we refer the reader to Ling et al. (2024); Bakman et al. (2024); Guerreiro et al. (2023); Zhang et al. (2023); Varshney et al. (2023); Luo et al. (2023); Ren et al. (2023); van der Poel et al. (2022); Wang et al. (2023).

# 3 Hallucination Detection Methodology

## 3.1 Problem Statement

We aim to model the binary classification problem of whether an LLM response contains a hallucination, which we define as any content that is nonfactual. To this end, we define a collection of binary classifiers, each of which map an LLM response $y_i \in \mathcal{Y}$, generated from prompt $x_i$, to a 'confidence score' between 0 and 1, where $\mathcal{Y}$ is the set of possible LLM outputs. We denote a hallucination classifier as $\hat{s} : \mathcal{Y} \to [0,1]$.

Given a classification threshold $\tau$, we denote binary hallucination predictions from the classifier as $\hat{h} : \mathcal{Y} \to \{0,1\}$. In particular, a hallucination is predicted if the confidence score is less than the threshold $\tau$:

$$\hat{h}(y_i; \cdot, \tau) = \mathbb{I}(\hat{s}(y_i; \cdot) < \tau), \tag{1}$$

where $\hat{s}$, and consequently $\hat{h}$, are conditioned on context variables including the prompt $x_i$ and, when applicable, additional responses generated from $x_i$.[5] Note that $\hat{h}(\cdot) = 1$ implies a hallucination is predicted. We denote the corresponding ground-truth label indicating whether the original response $y_i$ actually contains a hallucination as $h(y_i; \cdot)$. The function $h$ represents a grading process that compares a generated response $y_i$ to a correct reference answer $y_i^*$ for the same prompt $x_i$:

$$h(y_i; y_i^*, x_i) = \begin{cases} 1 & y_i \text{ is incorrect with respect to } y_i^* \\ 0 & \text{otherwise.} \end{cases} \tag{2}$$

When the correct answer $y_i^*$ is known, direct comparison yields the most accurate hallucination label. In deployment, however, $y_i^*$ is not available at generation time, so $h$ cannot be used as a real-time classifier. We therefore treat $h$ as an oracle ground truth used only offline for evaluation and, where applicable, tuning. Our framework approximates $h$ with $\hat{h}$ using uncertainty-based signals available at generation time, consistent with our closed-book setting and without access to $y_i^*$.

We adapt techniques from the uncertainty-quantification literature to compute response-level confidence for generation-time hallucination detection. Acknowledging the distinction between uncertainty quantification and hallucination detection, we study UQ-based confidence scorers for this purpose and do not distinguish between aleatoric and epistemic uncertainty. We transform and normalize outputs, if necessary, so that each score lies in $[0,1]$ with higher values indicating greater confidence.[6] Below, we detail these scorers.

## 3.2 Black-Box UQ Scorers

Black-box UQ scorers exploit variation in LLM responses to the same prompt to assess semantic consistency. For a given prompt $x_i$, these approaches involve generating $m$ candidate responses $\tilde{\mathbf{y}}_i = \{\tilde{y}_{i1}, ..., \tilde{y}_{im}\}$, using a non-zero temperature, from the same prompt and comparing these responses to the original response $y_i$. We provide detailed descriptions of each below.

**Exact Match Rate.** For LLM tasks that have a unique, closed-form answer, exact match rate can be a useful hallucination detection approach. Under this approach, an indicator function is used to score pairwise comparisons between the original response and the candidate responses. Given an original response $y_i$ and candidate responses $\tilde{\mathbf{y}}_i$, generated from prompt $x_i$, exact match rate (EMR) is computed as follows:

$$EMR(y_i; \tilde{\mathbf{y}}_i, x_i) = \frac{1}{m} \sum_{j=1}^{m} \mathbb{I}(y_i = \tilde{y}_{ij}). \tag{3}$$

---

[5]For notational simplicity, we use "·" as a placeholder here and explicitly write out conditioning variables in the scorer definitions that follow.

[6]Many scorers already output values in $[0,1]$ and therefore do not require normalization.

**Non-Contradiction Probability.** Non-contradiction probability (NCP) is a similar, but less stringent approach. NCP, a component of the BSDetector approach proposed by Chen & Mueller (2023), also conducts pairwise comparison between the original response and each candidate response. In particular, an NLI model is used to classify each pair $(y_i, \tilde{y}_{ij})$ as *entailment*, *neutral*, or *contradiction* and contradiction probabilities are saved. NCP for original response $y_i$ is computed as the average NLI-based non-contradiction probability across pairings with all candidate responses:

$$NCP(y_i; \tilde{\mathbf{y}}_i, x_i) = 1 - \frac{1}{m}\sum_{j=1}^{m} \frac{\eta(y_i, \tilde{y}_{ij}) + \eta(\tilde{y}_{ij}, y_i)}{2} \tag{4}$$

Above, $\eta(y_i, \tilde{y}_{ij})$ denotes the contradiction probability of $(y_i, \tilde{y}_{ij})$ estimated by the NLI model.[7] Following Chen & Mueller (2023) and Farquhar et al. (2024), we use `microsoft/deberta-large-mnli` for our NLI model.

**BERTScore.** Another approach for measuring text similarity between two texts is BERTScore (Zhang et al., 2020). Let a tokenized text sequence be denoted as $\mathbf{t} = \{t_1, ... t_L\}$ and the corresponding contextualized word embeddings as $\mathbf{E} = \{\mathbf{e}_1, ..., \mathbf{e}_L\}$, where $L$ is the number of tokens in the text. The BERTScore precision and recall scores between two tokenized texts $\mathbf{t}, \mathbf{t}'$ are respectively defined as follows:

$$BertP(\mathbf{t}, \mathbf{t}') = \frac{1}{|\mathbf{t}|}\sum_{t\in\mathbf{t}} \max_{t'\in\mathbf{t}'} \mathbf{e}\cdot\mathbf{e}'; \quad BertR(\mathbf{t}, \mathbf{t}') = \frac{1}{|\mathbf{t}'|}\sum_{t'\in\mathbf{t}'} \max_{t\in\mathbf{t}} \mathbf{e}\cdot\mathbf{e}' \tag{5}$$

where $e, e'$ respectively correspond to $t, t'$. We compute our BERTScore confidence (BSC) as follows:

$$BSC(y_i; \tilde{\mathbf{y}}_i, x_i) = \frac{1}{m}\sum_{j=1}^{m} 2\frac{BertP(y_i, \tilde{y}_{ij})BertR(y_i, \tilde{y}_{ij})}{BertP(y_i, \tilde{y}_{ij}) + BertR(y_i, \tilde{y}_{ij})}, \tag{6}$$

i.e. the average BERTScore F1 score across pairings of the original response with all candidate responses.

**Normalized Cosine Similarity.** Normalized cosine similarity (NCS) leverages a sentence transformer to map LLM outputs to an embedding space and measure similarity using those sentence embeddings. Let $V : \mathcal{Y} \to \mathbb{R}^d$ denote the sentence transformer, where $d$ is the dimension of the embedding space. We define NCS as the average cosine similarity across pairings of the original response with all candidate responses, normalized by dividing by 2 and adding $\frac{1}{2}$:

$$NCS(y_i; \tilde{\mathbf{y}}_i, x_i) = \frac{1}{2m}\sum_{j=1}^{m} \frac{\mathbf{V}(y_i)\cdot\mathbf{V}(\tilde{y}_{ij})}{\|\mathbf{V}(y_i)\|\|\mathbf{V}(\tilde{y}_{ij})\|} + \frac{1}{2}. \tag{7}$$

**Normalized Semantic Negentropy.** Semantic entropy (SE), proposed by Farquhar et al. (2024), exploits variation in multiple responses to compute a measure of response volatility. The SE approach clusters responses by mutual entailment and, like the NCP scorer, relies on an NLI model. However, in contrast to the aforementioned black-box UQ scorers, semantic entropy does not distinguish between an original response and candidate responses. Instead, it computes a single metric value on a list of responses generated from the same prompt. We consider the discrete version of SE, defined as follows:

$$SE(y_i; \tilde{\mathbf{y}}_i, x_i) = -\sum_{C\in\mathcal{C}} P(C|y_i, \tilde{\mathbf{y}}_i)\log P(C|y_i, \tilde{\mathbf{y}}_i), \tag{8}$$

where $P(C|y_i, \tilde{\mathbf{y}}_i)$ denotes the probability a randomly selected response $y \in \{y_i, \tilde{y}_{i1}, ..., \tilde{y}_{im}\}$ belongs to cluster $C$, and $\mathcal{C}$ denotes the full set of clusters of $\{y_i, \tilde{y}_{i1}, ..., \tilde{y}_{im}\}$.[8] To ensure that we have a normalized

---

[7]We note that NLI is an asymmetric measure. Manakul et al. (2023) propose a one-directional variant of NCP. We follow Chen & Mueller (2023) in using the bidirectional formulation to allow for more flexibility in detecting contradictions.

[8]If token probabilities of the LLM responses are available, the values of $P(C|y_i, \tilde{\mathbf{y}}_i)$ can be instead estimated using mean token probability. However, unlike the discrete case, this version of semantic entropy is unbounded and hence does not lend itself well to normalization.

confidence score with $[0, 1]$ support and with higher values corresponding to higher confidence, we implement the following normalization to arrive at *Normalized Semantic Negentropy* (NSN):

$$NSN(y_i; \tilde{\mathbf{y}}_i, x_i) = 1 - \frac{SE(y_i; \tilde{\mathbf{y}}_i, x_i)}{\log(m+1)}, \tag{9}$$

where $\log(m+1)$ is included to normalize the support.

### 3.3  White-Box UQ Scorers

White-box UQ scorers leverage token probabilities of the LLM's generated response to quantify uncertainty. We define two white-box UQ scorers below.

**Length-Normalized Token Probability.** Let the tokenization of LLM response $y_i$ be denoted as $\{t_1, ..., t_{L_i}\}$, where $L_i$ denotes the number of tokens the response. Length-normalized token probability (LNTP) computes a length-normalized analog of joint token probability:

$$LNTP(y_i; x_i) = \prod_{t \in y_i} p_t^{\frac{1}{L_i}}, \tag{10}$$

where $p_t$ denotes the token probability for token $t$.[9] Note that this score is equivalent to the geometric mean of token probabilities for response $y_i$.

**Minimum Token Probability.** Minimum token probability (MTP) uses the minimum among token probabilities for a given responses as a confidence score:

$$MTP(y_i; x_i) = \min_{t \in y_i} p_t, \tag{11}$$

where $t$ and $p_t$ follow the same definitions as above.

### 3.4  LLM-as-a-Judge Scorers

We employ an LLM-as-a-Judge scorer $J(y_i; x_i)$ as an additional method for obtaining response-level confidence scores. In this approach, we concatenate a question-response pair and pass it to an LLM with a carefully constructed instruction prompt that directs the model to evaluate the correctness of the response. We adapt our instruction prompt from Xiong et al. (2024), instructing the LLM to score responses on a 0-100 scale, where a higher score indicates a greater certainty that the provided response is correct. These scores are then normalized to a 0-to-1 scale to maintain consistency with our other confidence scoring methods. The complete prompt template is provided in Appendix A.

### 3.5  Ensemble Scorer

We introduce a tunable ensemble approach for hallucination detection. Specifically, our ensemble is a weighted average of $K$ binary classifiers: $\hat{s}_k : \mathcal{Y} \rightarrow [0, 1]$ for $k = 1, ..., K$. As several of our ensemble components exploit variation in LLM responses to the same prompt, our ensemble is conditional on $(\tilde{\mathbf{y}}_i, \mathbf{w})$, where $\mathbf{w}$ denote the ensemble weights. For original response $y_i$, we can write our ensemble classifier as follows:

$$\hat{s}(y_i; \tilde{\mathbf{y}}_i, x_i, \mathbf{w}) = \sum_{k=1}^{K} w_k \hat{s}_k(y_i; \tilde{\mathbf{y}}_i, x_i), \tag{12}$$

where $\mathbf{w} = (w_1, ..., w_K), \sum_{k=1}^{K} w_k = 1$, and $w_k \in [0, 1]$ for $k = 1, ..., K$.[10]

---

[9]Although it is not reflected in our notation, the probability for a given token is conditional on the preceding tokens.

[10]Note that although we write each classifier to be conditional on the set of candidate responses, some of the classifiers depend only on the original response.

Tuning the ensemble requires a sample of LLM responses $y_1, ..., y_n$ to a set of $n$ prompts, a set of $K$ confidence scores for each response $\{s_1(y_i; \tilde{\mathbf{y}}_i, x_i), ..., s_K(y_i; \tilde{\mathbf{y}}_i, x_i)\}_{i=1}^N$, and corresponding binary hallucination indicators $h(y_1; y_1^*, x_1), ..., h(y_n; y_n^*, x_n)$.[11] Given a classification objective function, the ensemble weights $\mathbf{w}$ can be tuned with an optimization routine.[12] If the objective is threshold-agnostic, the weights and threshold $\tau$ can be tuned sequentially. For a threshold-dependent objective, the weights and threshold can be tuned jointly. Because scorer performance depends on both the dataset and the underlying LLM, the ensemble weights are tuned per use case (chosen LLM and dataset), and the resulting weights are intended for in-domain deployment. See Appendix B for more details on ensemble tuning.

## 4 Experiments

### 4.1 Experiment Setup

We conduct a series of experiments to assess the hallucination detection performance of the various scorers. To accomplish this, we leverage a set of publicly available benchmark datasets that contain questions and answers. To ensure that our answer format has sufficient variation, we use two benchmarks with numerical answers (*GSM8K* (Cobbe et al., 2021) and *SVAMP* (Patel et al., 2021)), two with multiple-choice answers (*CSQA* (Talmor et al., 2022) and *AI2-ARC* (Clark et al., 2018)), and two with open-ended text answers (*PopQA* (Mallen et al., 2023) and *NQ-Open* (Lee et al., 2019)).

We sample 1000 questions for each of the six benchmarks. For each question, we generate an original response and $m = 15$ candidate responses using four LLMs: GPT-4o (OpenAI), GPT-4o-mini (OpenAI), Gemini-2.5-Flash (Google), and Gemini-2.5-Flash-Lite (Google).[13] Candidate responses are generated using a temperature of 1.0. For each response, we use the corresponding candidate responses generated by the same LLM to compute the full suite of black-box UQ scores. We extract the log-probabilities from the generated responses to compute white-box UQ scores. Lastly, we compute LLM-as-a-Judge scores for each response using the same four LLMs. We evaluate the hallucination detection performance of all individual scorers as well as our ensemble scorer for each of the benchmarks using various metrics, with grading processes detailed in Appendix C. All scores are computed using this paper's companion toolkit, `uqlm`.[14]

### 4.2 Experiment Results

**Threshold-Agnostic Evaluation** To assess the performance of the scorers as hallucination classifiers, we evaluate the performance of the confidence scores in a threshold-agnostic fashion.[15] Under this setting, we use the AUROC-optimized ensemble weights and compute the ensemble's AUROC using 5-fold cross-validation.[16] The final reported AUROC is obtained by averaging the AUROC values across the five holdout sets.

Figure 1 presents the AUROC scores for all scorers across the 24 LLM-dataset scenarios, while Table 1 highlights the best-performing scorer for each scenario. The AUROC values for the scenario-specific best scorers range from 0.729 for GPT-4o responses on NQ-Open (Ensemble) to 0.986 for Gemini-2.5-Flash-Lite responses on GSM8K (Ensemble). Overall, the top-performing scorers for each scenario exhibit strong hallucination detection performance, with AUROC values greater than 0.8 for 19 out of 24 scenarios. However, some scorers perform no better than or only negligibly better than a random classifier in hallucination detection in certain scenarios, such as the GPT-4o-mini LLM judge for all four GSM8K scenarios.

---

[11]Note that $h(y_1; y_1^*, x_1), ..., h(y_n; y_n^*, x_n)$ serve as 'ground truth' labels in the classification objective function.

[12]We use `optuna` (Akiba et al., 2019) for optimization with default settings (more details available here).

[13]We use a large number of candidate responses ($m = 15$) to ensure robust comparisons across black-box scorers. In practice, fewer candidates can be used. For an experimental evaluation of the impact of $m$ on performance, refer to Appendix D.

[14]Using an `n1-standard-16` machine (16 vCPU, 8 core, 60 GB memory) with a single NVIDIA T4 GPU attached, our experiments took approximately 0.5-3 hours per LLM-dataset combination to complete.

[15]In our experiments, we label 'correct' LLM responses as 1 and 'incorrect' responses as 0.

[16]Scorer performance depends on both the dataset and the LLM, so we tune the ensemble for a specific LLM–dataset pair. Our experiments reflect this in-domain setup; we do not evaluate out-of-distribution generalization across datasets or domains.

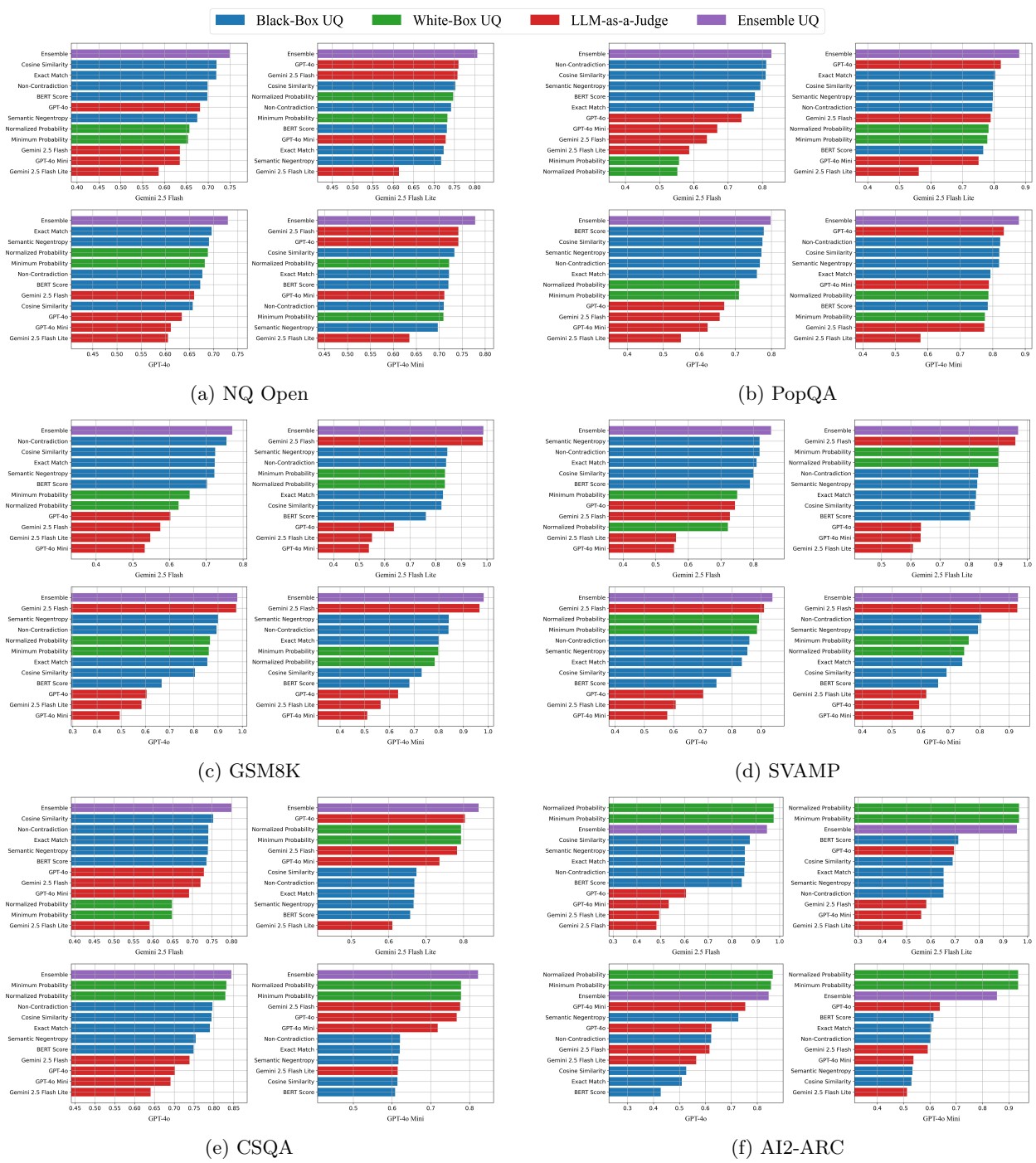

Figure 1: Scorer-Specific AUROC Scores for Hallucination Detection by LLM and Dataset (Higher is Better)

When comparing AUROC values across scorers, we find our ensemble scorer outperforms its individual components in 20 out of 24 scenarios, demonstrating the benefits of use-case-specific optimization. The rankings of individual scorers, however, vary significantly across scenarios, with LLM-as-a-Judge, black-box, and white-box methods achieving top non-ensemble performance in 11, 7, and 6 scenarios respectively. NLI-based scorers (NSN and NCP) exhibit strong performance, achieving the highest AUROC in 13 out of 24 scenarios among black-box scorers. Among LLM judges, GPT-4o and Gemini-2.5-Flash consistently outperform their smaller counterparts (GPT-4o-mini and Gemini-2.5-Flash-Lite), with GPT-4o ranking as

the highest-performing judge in 13 out of 24 scenarios and Gemini-2.5-Flash in 10 out of 24 scenarios. Notably, Gemini-2.5-Flash tended to be the highest performing LLM judge on the math benchmarks (6 out of 8 scenarios), and GPT-4o tended to be the highest performing judge on the short-answer benchmarks (6 out of 8 scenarios). Finally, the two white-box scorers perform approximately equally, with very similar AUROC scores in the vast majority of scenarios.

Table 1: Hallucination Detection AUROC (Higher is Better): Best-Performing Scorer by LLM and Dataset

| Model | Metric | NQ-Open | PopQA | GSM8K | SVAMP | CSQA | AI2-ARC |
|---|---|---|---|---|---|---|---|
| Gem.-2.5-Flash | AUROC | 0.749 | 0.826 | 0.771 | 0.854 | 0.800 | 0.975 |
| | Best Scorer | Ensemble | Ensemble | Ensemble | Ensemble | Ensemble | LNTP |
| Gem.-2.5-Flash-Lite | AUROC | 0.806 | 0.880 | 0.986 | 0.968 | 0.840 | 0.965 |
| | Best Scorer | Ensemble | Ensemble | Ensemble | Ensemble | Ensemble | LNTP |
| GPT-4o | AUROC | 0.729 | 0.798 | 0.979 | 0.938 | 0.844 | 0.860 |
| | Best Scorer | Ensemble | Ensemble | Ensemble | Ensemble | Ensemble | LNTP |
| GPT-4o-Mini | AUROC | 0.778 | 0.880 | 0.982 | 0.930 | 0.822 | 0.935 |
| | Best Scorer | Ensemble | Ensemble | Ensemble | Ensemble | Ensemble | LNTP |

**Threshold-Optimized Evaluation.** We evaluate the various scorers using a threshold-dependent metric (F1-score). To compute our ensemble scores in this setting, we jointly optimize the ensemble weights and threshold using F1-score as the objective function, as outlined in Appendix B. To ensure robust evaluations, we compute the scorer-specific F1-scores using 5-fold cross-validation. For each individual scorer, we select the F1-optimal threshold using grid search on the tuning set and compute F1-score on the holdout set. We report the final F1-score for each scorer as the average across holdout sets.

Table 2: Hallucination Detection F1-Scores (Higher is Better): Best-Performing Scorer by LLM and Dataset

| Model | Metric | NQ-Open | PopQA | GSM8K | SVAMP | CSQA | AI2-ARC |
|---|---|---|---|---|---|---|---|
| Gem.-2.5-Flash | Precision | 0.597 | 0.688 | 0.950 | 0.968 | 0.856 | 0.985 |
| | Recall | 0.907 | 0.890 | 0.999 | 0.999 | 0.986 | 0.987 |
| | F1-Score | 0.718 | 0.774 | 0.974 | 0.983 | 0.916 | 0.986 |
| | Best Scorer | Ensemble | Ensemble | Ensemble | Ensemble | Ensemble | NSN |
| Gem.-2.5-Fl.-Lt. | Precision | 0.587 | 0.680 | 0.960 | 0.977 | 0.863 | 0.968 |
| | Recall | 0.838 | 0.864 | 0.980 | 0.976 | 0.973 | 0.997 |
| | F1-Score | 0.688 | 0.759 | 0.969 | 0.977 | 0.915 | 0.982 |
| | Best Scorer | Ensemble | Ensemble | Ensemble | Gem.-2.5-Fl. | Ensemble | Gem.-2.5-Fl. |
| GPT-4o | Precision | 0.649 | 0.727 | 0.958 | 0.978 | 0.853 | 0.987 |
| | Recall | 0.917 | 0.880 | 0.987 | 0.990 | 0.972 | 1.000 |
| | F1-Score | 0.756 | 0.795 | 0.972 | 0.984 | 0.909 | 0.993 |
| | Best Scorer | Ensemble | Ensemble | Ensemble | Gem.-2.5-Fl. | NCP | Gem.-2.5-Fl. |
| GPT-4o-Mini | Precision | 0.616 | 0.700 | 0.906 | 0.969 | 0.840 | 0.952 |
| | Recall | 0.867 | 0.876 | 0.978 | 0.988 | 0.991 | 0.998 |
| | F1-Score | 0.718 | 0.776 | 0.940 | 0.978 | 0.909 | 0.975 |
| | Best Scorer | Ensemble | Ensemble | Ensemble | Ensemble | Ensemble | Gem.-2.5-Fl. |

Figure 2 displays the F1-scores for each scorer, while Table 2 summarizes the precision, recall, and F1 metrics for the top-performing scorer across all 24 evaluation scenarios. The results are largely consistent with the AUROC experiments. The ensemble scorer outperforms its individual components in most scenarios, achieving highest F1-score in 17 out of 24 scenarios. Similar to the AUROC experiments, the NLI-based scorers (NSN and NCP) often outperform other black-box scorers (18 out of 24 scenarios), consistent with the findings of previous studies (Kuhn et al., 2023; Manakul et al., 2023; Lin et al., 2024; Farquhar et al., 2024). Once again, the Gemini-2.5-Flash judge outperforms other LLM judges on the math benchmarks (8

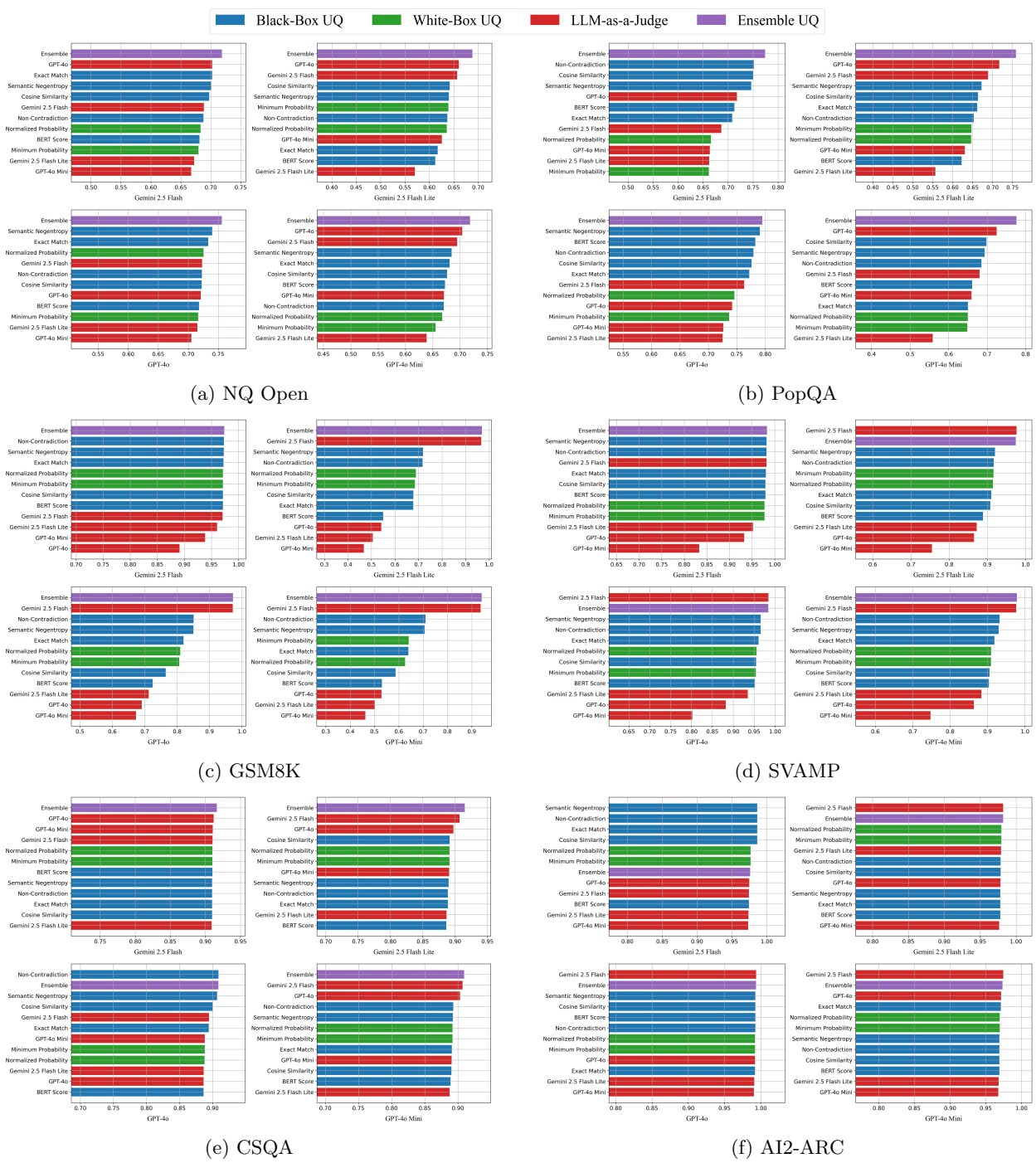

Figure 2: Scorer-Specific F1-Scores for Hallucination Detection by LLM and Dataset (Higher is Better)

out of 8 scenarios), while the GPT-4o judge outperforms other LLM judges on the short answer benchmarks (6 out of 8 scenarios). Interestingly, the strong LLM judge performance of Gemini-2.5-Flash on the math benchmarks and GPT-4o on the short answer benchmarks are consistent with their respective strengths when generating answers as the original LLM, where they achieved highest accuracy among the four LLMs on those same benchmarks (see Figure 3). Lastly, the two white-box scorers perform approximately equally, as in the AUROC experiments.

**Filtered Accuracy@$\tau$.** Lastly, we compute model accuracy on the subset of LLM responses having confidence scores exceeding a specified threshold $\tau$. We refer to this metric as *Filtered Accuracy@$\tau$*. Since the LLM accuracy depends on the choice of the threshold $\tau$, we repeat the calculation for $\tau = 0, 0.1, ..., 0.9$. Note that accuracy at $\tau = 0$ uses the full sample without score-based filtering.

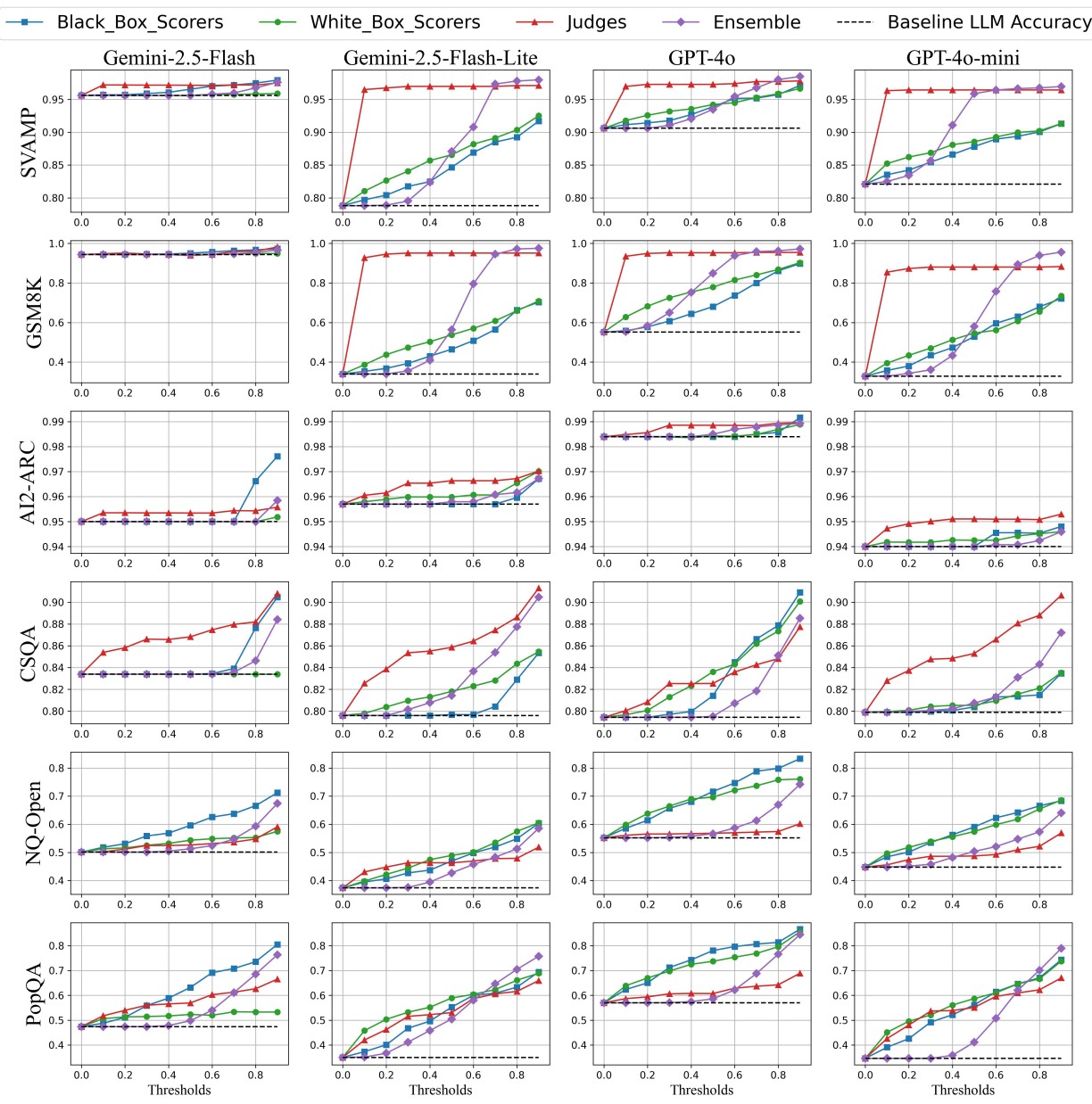

Figure 3: Filtered LLM Accuracy vs. Confidence Threshold (Top per Scorer Type)

Figure 3 presents the Filtered Accuracy@$\tau$ for the highest performing white-box, black-box, LLM-as-a-Judge, and ensemble scorers. Across all scenarios, response filtering with the highest-performing scorers leads to an approximately monotonic increase in LLM accuracy as the threshold increases. For example, when filtering Gemini-2.5-Flash-Lite responses to PopQA questions using the leading white-box scorer, accuracy improves dramatically from a baseline of 0.35 to 0.61 at $\tau = 0.6$. Similarly, with GPT-4o responses on GSM8K, filtering with the ensemble scorer achieves 0.93 accuracy at $\tau = 0.6$, substantially higher than the baseline accuracy of 0.55.

## 5   Discussion

**Choosing Among Scorers.**   Choosing the right confidence scorer for an LLM system depends on several factors, including API support, latency requirements, LLM behavior, and the availability of graded datasets. If the API supports access to token probabilities in LLM generations, white-box scorers can be implemented without adding latency or generation costs.[17]  However, if the API does not provide access to token probabilities, black-box scorers and LLM-as-a-Judge may be the only feasible options. When choosing among black-box and LLM-as-a-Judge scorers, latency requirements are a key consideration. For low-latency applications, practitioners should avoid higher-latency black-box scorers such as NLI-based scorers (NSN and NCP), opting instead for faster black-box scorers or LLM-as-a-Judge. If latency is not a concern, any of the black-box scorers may be suitable.

For LLM-as-a-Judge implementations, our findings reveal that an LLM's accuracy on a specific dataset positively relates to its ability to judge responses to questions from that same dataset, providing a practical criterion for judge selection. Finally, if a graded dataset is available, practitioners can tune an ensemble of various confidence scores to improve hallucination detection, as detailed in Appendix B. Our experiments demonstrate that a tuned ensemble can potentially provide more accurate confidence scores than individual scorers. By considering these factors and choosing the right confidence score, practitioners can improve the performance of their LLM system.

**Using Confidence Scores.**   In practice, practitioners may wish to use confidence scores for various purposes. First, practitioners can use our confidence scores for response filtering, where responses with low confidence are blocked, or 'targeted' human-in-the-loop, where low-confidence responses are selected for manual review. Our experimental evaluations of Filtered Accuracy@$\tau$ demonstrate the efficacy of these approaches, illustrating notable improvements in LLM accuracy when low-confidence responses are filtered out. Note that selecting a confidence threshold for flagging or blocking responses will depend on the scorer used, the dataset being evaluated, and stakeholder values (e.g., relative cost of false negatives vs. false positives).

Alternatively, confidence scores can be used for pre-deployment diagnostics, providing practitioners insights into the types of questions on which their LLM is performing worst. Findings from this type of exploratory analysis can inform strategies for improvements, such as further prompt engineering. Overall, the scorers included in our framework and toolkit provide practitioners with an actionable way to improve response quality, optimize resource allocation, and mitigate risks.

We note two important ethical considerations related to using confidence scores. First, a confidence score reflects model uncertainty rather than ground-truth-based correctness, and high scores can induce over-reliance in high-risk settings. We therefore caution against using confidence scores as decision guarantees in domains such as medicine, law, or finance without external human review and domain-specific safeguards. Second, we encourage monitoring for distributional fairness and content diversity, including per-segment error-rate evaluation, and, where applicable, safeguards that prevent systematic removal of accurate responses for particular groups.

**Limitations and Future Work.**   We note a few important limitations of this work. First, although our experiments leverage six question-answering benchmark datasets spanning three types of questions, we acknowledge that our findings on scorer-specific performance may not generalize to other types of questions.[18] For example, long-form generation (such as summarization) can mix true and false claims within a single output and may require claim-level decomposition to obtain useful uncertainty signals. Likewise, correctness in code generation depends on syntax and execution, so black-box consistency measurements will likely require alternative methods.

Second, while we conduct experiments using four LLMs, performance of the various scorers may differ for other LLMs. Note that for different LLMs, differences in token probability distributions will impact the

---

[17]The white-box scorers we consider in this work require only a single generation per prompt. However, sampling-based white-box methods also exist. See, for example, Kuhn et al. (2023); Scalena et al. (2025); Vashurin et al. (2025b;a); Qiu & Miikkulainen (2024).

[18]For practical reasons, we selected benchmark datasets containing questions that could be graded computationally.

behavior of white-box scorers, and the degree of variation in responses to the same prompt will affect the performance of black-box scorers. LLM-as-a-Judge performance may vary significantly depending on the choice of LLM and instruction prompts used.

Additional limitations relate to the ensemble approach. First, while the ensemble approach is designed to optimize in-domain performance, we have not evaluated out-of-distribution generalization, including cross-dataset transfer of learned weights; we leave this to future work. Second, our experiments consider only linear ensembles. Future work should examine non-linear ensembling and its impact on hallucination detection, e.g., monotonic generalized additive models that retain interpretability while capturing modest non-linear effects, mixture-of-experts, and tree-based ensembles. These may outperform linear averaging when scorer signals interact or exhibit non-linear relationships. Practical trade-offs to study include latency, overfitting risk on small graded sets, and the need for interpretability in high-stakes deployments. Third, tuning the ensemble requires a graded dataset. For tasks with trivial grading (e.g., arithmetic, multiple-choice, short-answer), where there are well-defined, easy-to-recognize correct answers and comparisons with generated text can be computed automatically, this can be done by sampling a representative set of questions, generating responses, computing the selected UQ scores, and using the automatically derived labels to fit the weights. For tasks with non-trivial grading (e.g., summarization), practitioners can begin with a small human-graded dataset (e.g., a few hundred items); additional labels can then be added incrementally from production logs to further optimize the weights and improve ensemble performance.

## 6 Conclusions

In this paper, we detail a framework for closed-book hallucination detection comprised of various black-box UQ, white-box UQ, and LLM-as-a-Judge scorers. To ensure standardized outputs of the scorers, we transform and normalize scorers (if necessary) such that all outputs range from 0 to 1, with higher scores indicating greater confidence in an LLM response. These response-level confidence scores can be used for generation-time hallucination detection across a wide variety of LLM use cases. Additionally, we introduce a novel, ensemble-based approach that leverages an optimized weighted average of any combination of individual confidence scores. Importantly, the extensible nature of our ensemble means that practitioners can include additional scorers as new methods become available.

Our experimental evaluation of UQ-based scorers offers clear guidance for practitioners. Ensemble approaches consistently outperform individual methods for hallucination detection, with our findings strongly supporting use-case-specific customization rather than one-size-fits-all solutions. For those without token-probability access, NLI-based approaches typically provide the best black-box performance. Importantly, gains from sampling additional responses diminish as the number of candidate responses increases, offering a practical deployment guideline that balances effectiveness with computational efficiency. Finally, our analysis revealed that a model's accuracy on a specific dataset positively relates to its ability to judge responses to questions from that same dataset, providing a practical criterion for judge selection in evaluation frameworks.

The `uqlm` library implements all scorers presented and evaluated in this work. The repository is actively maintained. We welcome issues and pull requests and will continue to update integrations, add new scorers as research advances, and refresh examples as model and provider APIs change.

## Acknowledgements

We wish to thank David Skarbrevik, Piero Ferrante, Xue (Crystal) Gu, Blake Aber, Viren Bajaj, Ho-Kyeong Ra, Zeya Ahmad, Matthew Churgin, Saicharan Sirangi, and Erik Widman for their helpful suggestions.

## Conflict of Interest

DB and MSC are employed and receive stock and equity from CVS Health® Corporation.

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

# A    LLM-as-a-Judge Prompt

Our LLM-as-a-Judge scorer used the following instruction prompt:

> Question: [question], Proposed Answer: [answer].
>
> How likely is the above answer to be correct? Analyze the answer and give your confidence in this answer between 0 (lowest) and 100 (highest), with 100 being certain the answer is correct, and 0 being certain the answer is incorrect. THE CONFIDENCE RATING YOU PROVIDE MUST BE BETWEEN 0 and 100. ONLY RETURN YOUR NUMERICAL SCORE WITH NO SURROUNDING TEXT OR EXPLANATION.
>
> \# Example 1
> \#\# Data to analyze
> Question: Who was the first president of the United States?, Proposed Answer: Benjamin Franklin.
>
> \#\# Your response
> 4 (highly certain the proposed answer is incorrect)
>
>
> \# Example 2
> \#\# Data to analyze
> Question: What is 2+2?, Proposed Answer: 4
>
> \#\# Your response
> 99 (highly certain the proposed answer is correct)

To ensure a normalized confidence score consistent with the other scorers, we normalize the value returned by the LLM judge to be between 0 and 1. The capitalization and repeated instructions, inspired by Wang et al. (2024), are included to ensure the LLM correctly follows instructions.

# B    Ensemble Tuning

We outline a method for tuning ensemble weights for improved hallucination detection accuracy. This approach allows for customizable component-importance that can be optimized for a specific use case. In practice, tuning the ensemble weights requires having a 'graded' set of $n$ original LLM responses which indicate whether a hallucination is present in each response.[19] Given a set of $n$ prompts, denoted as $\mathbf{x}$:

$$\mathbf{x} = \begin{pmatrix} x_1 \\ x_2 \\ \vdots \\ x_n \end{pmatrix}, \tag{13}$$

denote corresponding correct reference answers as $\mathbf{y}^*$,

$$\mathbf{y}^* = \begin{pmatrix} y_1^* \\ y_2^* \\ \vdots \\ y_n^* \end{pmatrix}, \tag{14}$$

---

[19]Grading responses may be accomplished computationally for certain tasks, e.g. multiple choice questions. However, in many cases, this will require a human grader to manually evaluate the set of responses.

original LLM responses as $\mathbf{y}$,

$$\mathbf{y} = \begin{pmatrix} y_1 \\ y_2 \\ \vdots \\ y_n \end{pmatrix}, \tag{15}$$

and candidate responses across all prompts with the matrix $\tilde{\mathbf{Y}}$:

$$\tilde{\mathbf{Y}} = \begin{pmatrix} \tilde{\mathbf{y}}_1 \\ \tilde{\mathbf{y}}_2 \\ \vdots \\ \tilde{\mathbf{y}}_n \end{pmatrix} = \begin{pmatrix} \tilde{y}_{11} & \tilde{y}_{12} & \cdots & \tilde{y}_{1m} \\ \tilde{y}_{21} & \tilde{y}_{22} & \cdots & \tilde{y}_{2m} \\ \vdots & \vdots & \ddots & \vdots \\ \tilde{y}_{n1} & \tilde{y}_{n2} & \cdots & \tilde{y}_{nm} \end{pmatrix}. \tag{16}$$

Analogously, we denote the vectors of ensemble confidence scores, binary ensemble hallucination predictions, and corresponding ground truth values respectively as

$$\hat{\mathbf{s}}(\mathbf{y}; \tilde{\mathbf{Y}}, \mathbf{x}, \mathbf{w}) = \begin{pmatrix} \hat{s}(y_1; \tilde{\mathbf{y}}_1, x_1, \mathbf{w}) \\ \hat{s}(y_2; \tilde{\mathbf{y}}_2, x_2, \mathbf{w}) \\ \vdots \\ \hat{s}(y_n; \tilde{\mathbf{y}}_n, x_n, \mathbf{w}) \end{pmatrix}, \tag{17}$$

$$\hat{\mathbf{h}}(\mathbf{y}; \tilde{\mathbf{Y}}, \mathbf{x}, \mathbf{w}, \tau) = \begin{pmatrix} \hat{h}(y_1; \tilde{\mathbf{y}}_1, x_1, \mathbf{w}, \tau) \\ \hat{h}(y_2; \tilde{\mathbf{y}}_2, x_2, \mathbf{w}, \tau) \\ \vdots \\ \hat{h}(y_n; \tilde{\mathbf{y}}_n, x_n, \mathbf{w}, \tau) \end{pmatrix}, \tag{18}$$

and

$$\mathbf{h}(\mathbf{y}; \mathbf{y}^*, \mathbf{x}) = \begin{pmatrix} h(y_1; y_1^*, x_1) \\ h(y_2; y_2^*, x_2) \\ \vdots \\ h(y_n; y_n^*, x_n) \end{pmatrix}, \tag{19}$$

Modeling this problem as binary classification enables us to tune the weights of our ensemble classifier using standard classification objective functions. Following this approach, we consider two distinct strategies to tune ensemble weights $w_1, ..., w_K$: threshold-agnostic optimization and threshold-aware optimization.

**Threshold-Agnostic Weights Optimization.** Our first ensemble tuning strategy uses a threshold-agnostic objective function for tuning the ensemble weights. Given a set of $n$ prompts, corresponding original LLM responses and candidate responses, the optimal set of weights, $\mathbf{w}^*$, is the solution to the following problem:

$$\mathbf{w}^* = \arg\max_{\mathbf{w} \in \mathcal{W}} \mathcal{S}(\hat{\mathbf{s}}(\mathbf{y}; \tilde{\mathbf{Y}}, \mathbf{x}, \mathbf{w}), \mathbf{h}(\mathbf{y}; \mathbf{y}^*, \mathbf{x})), \tag{20}$$

where

$$\mathcal{W} = \{(w_1, ..., w_K) : \sum_{k=1}^{K} w_k = 1, w_k \geq 0 \ \forall \ k = 1, ..., K\} \tag{21}$$

is the support of the ensemble weights and $\mathcal{S}$ is a threshold-agnostic classification performance metric, such as area under the receiver-operator characteristic curve (AUROC).

After optimizing the weights, we subsequently tune the threshold using a threshold-dependent objective function. Hence, the optimal threshold, $\tau^*$, is the solution to the following optimization problem:

$$\tau^* = \arg\max_{\tau \in (0,1)} \mathcal{B}(\hat{\mathbf{h}}(\mathbf{y}; \tilde{\mathbf{Y}}, \mathbf{x}, \mathbf{w}^*, \tau), \mathbf{h}(\mathbf{y}; \mathbf{y}^*, \mathbf{x})), \tag{22}$$

where $\mathcal{B}$ is a threshold-dependent classification performance metric, such as F1-score.

**Threshold-Aware Weights Optimization.** Alternatively, practitioners may wish jointly optimize ensemble weights and classification threshold using the same objective. This type of optimization relies on a threshold-dependent objective. We can write this optimization problem as follows:

$$\mathbf{w}^*, \tau^* = \arg\max_{\mathbf{w} \in \mathcal{W}, \tau \in (0,1)} \mathcal{B}(\hat{\mathbf{h}}(\mathbf{y}; \tilde{\mathbf{Y}}, \mathbf{x}, \mathbf{w}, \tau), \mathbf{h}(\mathbf{y}; \mathbf{y}^*, \mathbf{x})), \tag{23}$$

where terms follow the same definitions as above.

## C  Response Grading

For each task-type, the LLM is instructed to output its response in a specific form. We therefore instantiate $h(\cdot)$ with task-specific graders aligned to that form. Unless noted, grading is case-insensitive and trims whitespace.[20]

**Math (numeric answer only).** For math questions, the LLM is provided the following instruction: "Return only the numerical answer with no additional text." Let $y_i$ be the model output. Let $\text{int}(y_i)$ extract the leading integer substring from $y_i$ if present; otherwise return $\varnothing$. Let $y_i^* \in \mathbb{Z}$ be the correct integer answer.

$$h_{\text{math}}(y_i; y_i^*) = \begin{cases} 0 & \text{if } \text{int}(y_i) = y_i^*, \\ 1 & \text{otherwise.} \end{cases}$$

Non-numeric or missing integer outputs are graded as incorrect.

**Multiple choice (letter only).** For multiple choice questions, the LLM is provided the following instruction: "Return only the letter of the response with no additional text or explanation." Let $\text{norm}_{mc}(y_i) \in \{A, B, C, D, E\}$ denote response $y_i$ after normalization to uppercase and trimming. Let $y_i^* \in \{A, B, C, D, E\}$ be the correct letter.

$$h_{\text{mc}}(y_i; y_i^*) = \begin{cases} 0 & \text{if } \text{norm}_{mc}(y_i) = y_i^*, \\ 1 & \text{otherwise.} \end{cases}$$

Any invalid response based on the provided instruction, i.e. $\text{norm}_{mc}(y_i) \notin \{A, B, C, D, E\}$, is graded as incorrect.

**Short-answer (keyword match).** Lastly, for short-answer questions, the LLM is instructed as follows: "Return only the answer as concisely as possible without providing an explanation." Let $\mathbf{y}_i^* = \{y_{i1}^*, \ldots, y_{iA}^*\}$ be the set of acceptable answers.[21] Define a normalization function $\text{norm}_{sa}(y_i)$ that lowercases and trims $y_i$. Let $\text{contains}(u, v)$ indicate that string $u$ contains string $v$ as a contiguous substring.

$$h_{\text{sa}}(y_i; \mathbf{y}_i^*) = \begin{cases} 0 & \text{if } \exists\, y_i^* \in \mathbf{y}_i^* \text{ s.t. } \text{contains}(\text{norm}_{sa}(y_i), \text{norm}_{sa}(y_i^*)), \\ 1 & \text{otherwise.} \end{cases}$$

If output includes additional text, grading is based on the presence of any acceptable answer string.

---

[20]These automatic graders may not be perfectly accurate in every instance; residual errors can arise when model outputs deviate from the specified response format. Their effectiveness therefore depends on instruction adherence, and they should be viewed as practical proxies for answer-key–based factual correctness.

[21]The short-answer datasets contain multiple versions of the same correct answer to allow for phrasing flexibility. For example, acceptable answers to the question "What is Tobias Lindholm's occupation?" include {"film director", "movie director", "director", "motion picture director", "screenwriter", "scenarist", "writer", "screen writer", "script writer", "scriptwriter"}.

# D   Additional Experiments: Number of Candidate Responses vs. Black-Box UQ Performance

To investigate the effect of number of candidate responses $m$ on the performance of black-box scorers, we re-compute all black-box confidence scores for $m = 1, 3, 5, 10, 15$ for each of our 24 LLM-dataset scenarios. We compute the scorer-specific AUROC value for each value of $m$ and evaluate the impact of number of candidate responses on hallucination detection performance. These results are depicted in Figure 4.

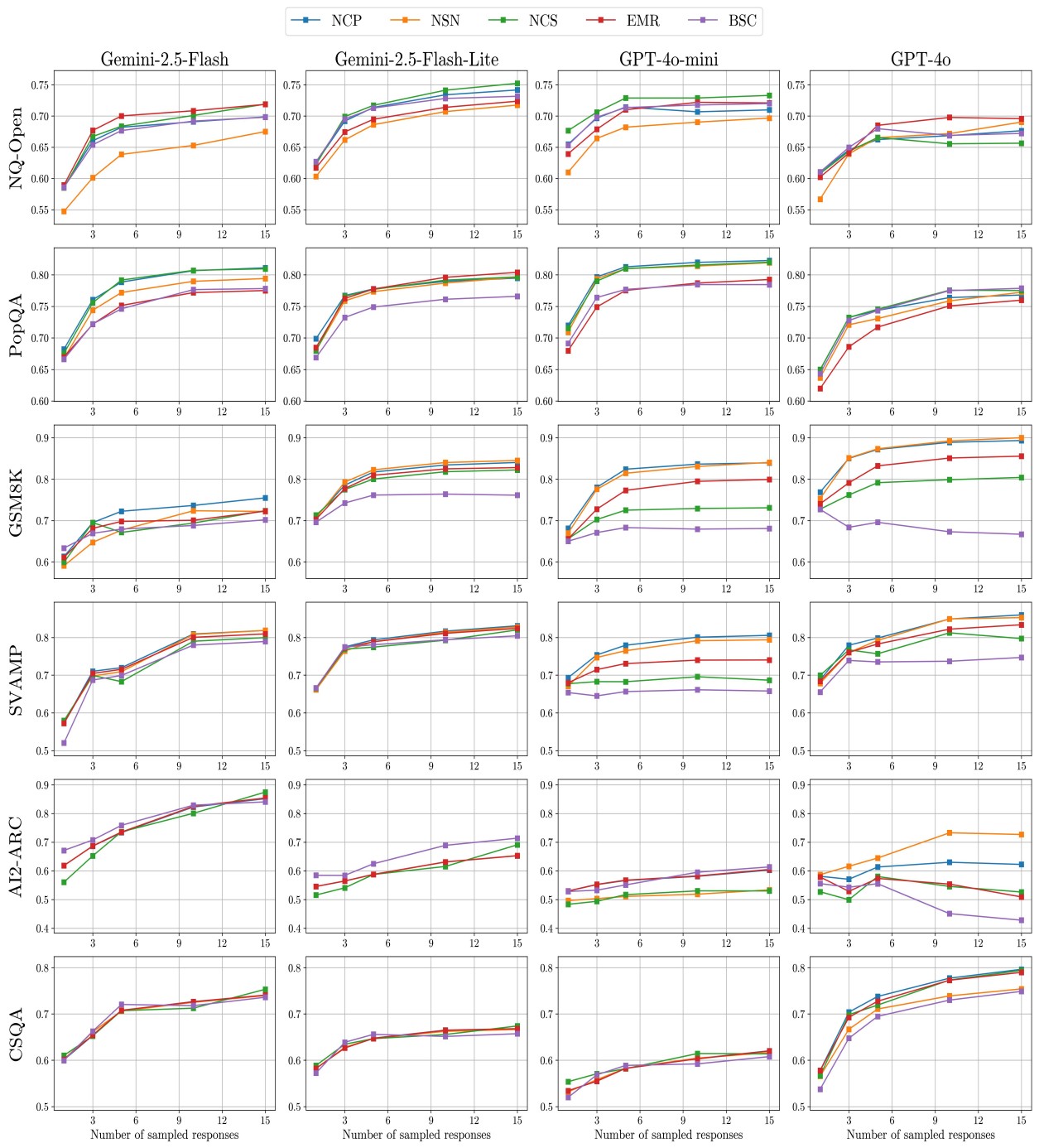

Figure 4: Hallucination Detection AUROC by Number of Sampled Responses

Overall, the results indicate that hallucination detection performance of the black-box scorers improves considerably with number of candidate responses $m$. For instance, hallucination detection AUROC of the various black-box scorers on GPT-4o responses on CSQA improve from 0.54-0.57 with $m = 1$ to 0.75-0.8 with $m = 15$. In particular, these performance improvements occur approximately monotonically with diminishing returns to higher $m$, consistent with findings from previous studies (Kuhn et al., 2023; Manakul et al., 2023; Lin et al., 2024; Farquhar et al., 2024). This general trend is consistent across all 24 scenarios, with only a few exceptions, notably BSC for GPT-4o responses on GSM8k, and BSC, NCS, and EMR for GPT-4o responses on AI2-ARC.

