# OpenReview forum: "Uncertainty Quantification for Language Models: A Suite of Black-Box, White-Box, LLM Judge, and Ensemble Scorers"
_TMLR — Accepted by TMLR_

### Review · Reviewer_XZVQ · 2025-09-09

**Summary Of Contributions:**

This paper introduces a practical framework for detecting LLM hallucinations in real-time without needing ground truth data. Its primary contribution is adapting various existing Uncertainty Quantification (UQ) techniques - specifically black-box, white-box, and LLM-as-a-Judge methods - and standardizing them into a unified confidence score. The authors then propose a tunable ensemble that combines these individual scorers as a weighted average. This allows a user to optimize the hallucination detector for their specific LLM and data. The authors provide an open-source Python toolkit for this system and experimentally compare their customized ensemble approach to the other scoring methods.

The framework's key strengths are its practicality (it is "zero-resource"), its flexibility, and its accessibility via the provided toolkit. Moreover, they test the different approaches in a standardized and clean experimental setup, explaining the pros and cons of each method. The main weaknesses are that all experiments were limited to question-answering tasks (so the findings may not generalize to other tasks succh as summarization) and that only a simple ensemble method (linear combination) was implemented.

**Audience:**

No

**Audience Explanation:**

Researchers and practitioners who aim to deploy LLMs, and who are interested in understanding hallucination rates, can learn about the different approaches outlined in this paper from their original publications. The method suggested by this paper- tuning an ensemble - is not particularly novel. This practice is common in many other contexts, and applying it specifically to hallucination detection is a natural step that others have likely already considered or implemented.

**Broader Impact Concerns:**

None.

**Claims And Evidence:**

Yes

**Claims Explanation:**

The paper’s central thesis is that its proposed tunable ensemble outperforms any single Uncertainty Quantification scorer. This claim is convincingly validated through extensive experiments, covering several different LLMs and several QA benchmarks. The results show that the ensemble achieved the highest hallucination detection performance in most of the scenarios.

**Requested Changes:**

Due to the fundamental lack of novelty, there are no feasible changes I can recommend.

---

> ### Author Response · Authors · 2025-10-08
>
> Thank you for the thoughtful review and for affirming that our core claims are accurately and convincingly supported across multiple LLMs and QA benchmarks.
>
> ### **On audience interest (per TMLR criteria).**
> While the ensemble itself is simple, and recognizing that TMLR does not include technical novelty in its acceptance criteria, we believe the contribution of this work is valuable to a portion of TMLR's readership because it delivers:
>  - **Zero-resource, generation-time detection**: The framework standardizes heterogeneous UQ signals into a 0 to 1 confidence that can be used immediately in real-time monitoring without external corpora or retrieval.
> - **A rigorous, standardized evaluation**: We provide a carefully controlled comparison of black-box, white-box, and judge-based scorers across multiple LLMs and benchmarks, creating a clear empirical reference that practitioners and researchers can rely on.
> - **Consistent performance gains from a tunable ensemble**: Despite its simplicity, the ensemble achieves the strongest results across most settings, offering a straightforward, reproducible recipe for improved detection performance.
> - **An open-source toolkit**: The code lowers adoption barriers, supports reproduction, and allows others to extend individual scorers or the ensemble, which is practically useful for deployment-focused readers.
>
> To further aid applicability, we expanded the Discussion (sec. 5) to outline future non-linear ensemble directions, to explain task-scope limitations for long-form and code generation, and to add concrete guidance on graded-data acquisition for tuning. These additions clarify how the community can extend and apply the framework beyond our current scope.
>
> We appreciate the opportunity to address these concerns and hope that both our explanation above and our revisions have successfully demonstrated the practical utility and evaluative clarity our work offers to researchers and practitioners in the TMLR community working on reliability and applied evaluation.

---

> > ### Comment · Reviewer_XZVQ · 2025-10-22
> > **Thanks for your reply**
> >
> > Thanks for drawing my attention to TMLR's novelty criteria.
> > I acknowledge that my initial assessment of the "audience interest" criterion was unduly influenced by a focus on novelty which, as you rightly point out, is not a primary criterion for TMLR. Therefore, I agree that your submission will be of interest to the TMLR readership.

---

### Review · Reviewer_DY7K · 2025-09-13

**Summary Of Contributions:**

I consider this paper to make a useful contribution to addressing the challenge of Large Language Model (LLM) reliability in practical settings.

A central aspect of the paper is its presentation of a framework for zero-resource hallucination detection at the time responses are generated by LLMs. This approach is relevant for real-time monitoring of LLM outputs in production, as it does not rely on external databases, ground truth texts, or internet access.

Within this framework, the authors have adapted various existing Uncertainty Quantification (UQ) techniques—including black-box UQ, white-box UQ, and LLM-as-a-Judge methods—and converted their outputs into standardized response-level confidence scores ranging from 0 to 1. This standardization helps in making different UQ methods more consistent and comparable.

The paper also introduces a tunable ensemble approach. This method allows for combining individual confidence scores through a weighted average, with the weights being adjustable based on a provided set of graded LLM responses. The intent here is to optimize hallucination detection for specific use cases. The ensemble is also designed to be extensible, allowing for the inclusion of new components as research progresses.

Finally, the paper offers practical guidance for practitioners on selecting appropriate scorers, considering factors like API support for token probabilities, latency requirements, and the LLM's response behavior. This work is complemented by an open-source Python toolkit, which provides implementations of the UQ methods discussed.

Strengths:
• Practical Orientation: The focus on zero-resource, generation-time hallucination detection directly addresses a need for improving LLM reliability in deployed systems, particularly in sensitive domains.
• Standardized Approach: The effort to standardize various UQ techniques into a 0-1 confidence score is a practical step, contributing to a more consistent way of assessing LLM output confidence.
• Tunable Ensemble: The tunable ensemble approach is a notable feature. It generally demonstrates improved performance over individual scorers in the experiments and allows for some customization based on specific use case needs.
• Empirical Testing and Accessibility: The paper includes experiments across different LLMs and benchmarks, providing some evidence for the framework's effectiveness. The provision of an open-source toolkit facilitates the adoption of these methods.

Weaknesses:
• Reliance on Existing UQ Concepts: While the framework and ensemble are presented as a synthesis, the underlying black-box, white-box, and LLM-as-a-Judge methods are adaptations of techniques previously described in literature.
• Limited Generalization of Task Types: The experimental validation primarily uses question-answering benchmarks. The authors note that the findings on scorer performance may not directly extend to other tasks, such as summarization or longer-form text generation.
• Ensemble Tuning Requires Graded Data: To effectively tune the ensemble, a "graded" dataset of LLM responses indicating hallucinations is necessary. This may still require manual effort in many real-world scenarios.
• Scope of Ensemble Exploration: The experiments were limited to linear ensembles. Further exploration of more complex ensembling techniques could potentially yield additional insights or performance gains.

**Audience:**

Yes

**Audience Explanation:**

I would assess this as yes.

The paper addresses a recognized challenge in LLM development: hallucination, particularly relevant for models deployed in high-stakes fields like healthcare and finance. Its proposed framework offers a method for zero-resource, real-time hallucination detection, which is a practical concern for monitoring LLM systems in production environments.

The tunable ensemble approach for combining various uncertainty quantification techniques seems to generally outperform individual methods and can be customized for specific use cases, which could appeal to researchers interested in model reliability and optimization.
The availability of an open-source toolkit for implementing these methods also makes the work more accessible and actionable for the community

**Broader Impact Concerns:**

The contribution here is largely positive in addressing LLM hallucination. Some general considerations though
1. Bias Amplification: If the UQ scorers or underlying LLMs have biases, filtering low-confidence responses could unfairly block accurate information for certain groups, thus amplifying existing societal biases.
2. Misinterpretation/Over-reliance: Confidence scores, especially in "high-stakes domains," could be misinterpreted as absolute truth or lead to over-reliance, potentially diminishing critical human oversight.
3. Information Access: Aggressive filtering based on confidence thresholds, which depend on "stakeholder values," might inadvertently narrow the diversity of information an LLM provides, even if the intent is to block hallucinations.

These points highlight ethical implications that need explicit discussion and mitigation strategies.

**Claims And Evidence:**

Yes

**Claims Explanation:**

In my view, the authors have undertaken a structured evaluation, and their findings appear to back their assertions.

Framework for Real-time, Zero-resource Detection:
The definitions provided for zero-resource (no external databases, ground truth, or internet access) and generation time (suitable for real-time monitoring) are clear. The experimental setup, where scores are computed on LLM outputs, implicitly confirms these operational parameters.

Adaptation and Standardization of UQ Techniques:
The submission claims to adapt existing black-box, white-box, and LLM-as-a-Judge UQ techniques, normalizing them to a 0-1 confidence score. The paper explicitly details how various methods like Exact Match Rate, Length-Normalized Token Probability, and LLM-as-a-Judge scores are calculated and, if necessary, transformed or normalized to fit this. This methodical approach provides clear support for the standardization claim.

Tunable Ensemble Performance:
A core claim is the proposal of a tunable ensemble approach that typically surpasses its individual components. The tuning methodology, involving threshold-agnostic or threshold-aware optimization with graded LLM responses, is clearly outlined. Empirically, the ensemble is shown to outperform its individual components in 18 out of 24 scenarios for AUROC and achieving highest F1-score in 16 out of 24 scenarios. This consistent performance across multiple scenarios, as presented in Tables 1 and 2, provides convincing evidence for its general superiority.

Practical Guidance and Toolkit: The python code uploaded is readable and support the experiments.

The evidence is accurate as it stems from a detailed experimental setup with specified LLMs, datasets, and metrics. It is convincing due to the breadth of evaluation scenarios and the consistent trends observed.

The acknowledgment of limitations, such as focusing on question-answering tasks and linear ensembles, also adds to the paper's credibility.

**Requested Changes:**

Here are my requested changes as feasible:

1. Clarify the "Zero-Resource" Scope: Please elaborate slightly on how the "zero-resource" claim specifically applies to the LLM-as-a-Judge component. A brief clarification on whether the judge LLM's own knowledge base impacts this classification would be helpful.
2. Enhance Guidance on Graded Data Acquisition: While the need for graded data for ensemble tuning is mentioned, expanding on practical strategies for obtaining or scaling these datasets, or discussing associated trade-offs, would make the tunable ensemble more accessible for real-world implementation.
3. Explain Task Scope Limitations: For the acknowledged limitation regarding generalization to summarization or other long-form tasks, a brief explanation of why the findings might not extend (e.g., different hallucination types, signal changes) would be beneficial.
4. Briefly Outline Future Ensemble Directions: Since only linear ensembles were explored, a short note on potential non-linear ensemble techniques for future work, and why they might offer further improvements, would be a valuable addition.

Thank you for your considerations here

---

> ### Author Response · Authors · 2025-10-08
> **Requested changes incorporated in updated manuscript**
>
> Thank you for the thoughtful, actionable suggestions. We have updated the manuscript accordingly.
>
> ### **Requested Changes**
> 1. **Clarify the "Zero-Resource" Scope**: In Sec. 3.4 (LLM-as-a-Judge) we added Footnote 7: "Our use of ``zero-resource'' means no external corpora, retrieval, ground-truth texts, or internet search access at scoring time. An LLM judge satisfies this because it operates only on $(x_i,y_i)$ and the instruction prompt, using its fixed internal parameters. We do not enable tools, browsing, code execution, or retrieval, nor do we supply auxiliary context beyond the question and response. A tool- or retrieval-augmented judge would fall outside our zero-resource scope. While we acknowledge that the judge’s internal knowledge can influence judgments, we do not consider this an external resource requirement."
> 2. **Enhance Guidance on Graded Data Acquisition**: In **Discussion** → **Limitations and Future Work** (Sec. 5) we added: "...tuning the ensemble requires a graded dataset. For tasks with trivial grading (e.g., arithmetic, multiple-choice, short-answer), where there are well-defined, easy-to-recognize correct answers and comparisons with generated text can be computed automatically, this can be done by sampling a representative set of questions, generating responses, computing the selected UQ scores, and using the automatically derived labels to fit the weights. For tasks with non-trivial grading (e.g., summarization), practitioners can begin with a small human-graded dataset (e.g., a few hundred items); additional labels can then be added incrementally from production logs to further optimize the weights and improve ensemble performance."
> 3. **Explain Task Scope Limitations**: In **Discussion → Limitations and Future Work (Sec. 5)** we added: "For example, long-form generation (such as summarization) can mix true and false claims within a single output and may require claim-level decomposition to obtain useful uncertainty signals. Likewise, correctness in code generation depends on syntax and execution, so black-box consistency measurements will likely require alternative methods."
> 4. **Briefly Outline Future Ensemble Directions**: In **Discussion → Limitations and Future Work (Sec. 5)** we added: "Future work should examine non-linear ensembling and its impact on hallucination detection, e.g., monotonic generalized additive models that retain interpretability while capturing modest non-linear effects, mixture-of-experts, and tree-based ensembles. These may outperform linear averaging when scorer signals interact or exhibit non-linear relationships. Practical trade-offs to study include latency, overfitting risk on small graded sets, and the need for interpretability in high-stakes deployments."
>
> ### **Edits from Broader Impact Concerns**
> We appreciate the points on bias amplification, over-reliance, and information access. In **Discussion** → **Using Confidence Scores** (Sec. 5) we added:
>
> "We note two important ethical considerations related to using confidence scores. First, a confidence score reflects model uncertainty rather than ground-truth-based correctness, and high scores can induce over-reliance in high-risk settings. We therefore caution against using confidence scores as decision guarantees in domains such as medicine, law, or finance without external human review and domain-specific safeguards. Second, we encourage monitoring for distributional fairness and content diversity, including per-segment error-rate evaluation, and, where applicable, safeguards that prevent systematic removal of accurate responses for particular groups."
>
> We hope these updates address your concerns and make the practical scope and responsible-use guidance clearer.

---

> > ### Comment · Reviewer_DY7K · 2025-10-15
> > **Thank you for incorporating the changes and effectively addressing the comments.**
> >
> > Thank you for your prompt response and effectively addressing the requested changes in the updated manuscript. I have reviewed the incorporated edits, and they substantially clarify the scope and practical implications of the work. I am satisfied with the inclusion of ethical considerations as well. I do not have any further comments pertaining to this paper.

---

### Review · Reviewer_BYp9 · 2025-10-07

**Summary Of Contributions:**

This paper addresses the problem of Hallucinations in Large Language Models (LLMs) by developing a comprehensive approach to uncertainty quantification (UQ). The work systematically investigates three distinct UQ techniques: black-box UQ, white-box UQ, and LLM-as-a-Judge, with each technique outputting a confidence score for the LLM's response. A key contribution is the introduction of an ensemble method that fuses these individual scores using a weighted average. The weights for this ensemble are specifically tuned to maximize the F1 score as a performance metric. The efficacy of this novel, combined method is thoroughly validated through extensive experiments on several standard LLM question-answering benchmarks.

**Audience:**

Yes

**Audience Explanation:**

Yes, the findings of this paper would be of significant interest to a substantial portion of TMLR's audience, as the work addresses one of the most critical and timely challenges in modern AI: Large Language Model (LLM) reliability and safety.

**Broader Impact Concerns:**

The work, while aimed at improving LLM reliability, introduces ethical concerns primarily related to the risk of misplaced trust and "confidence laundering." By generating a seemingly objective, quantitative confidence score through Uncertainty Quantification (UQ), the ensemble method risks convincing users and automated systems to over-rely on the LLM's output, especially when the score is high. If the UQ system is imperfectly calibrated, a high confidence score could falsely legitimize a residual hallucination or bias, thereby elevating the operational risk in sensitive applications like finance, law, or medicine. Consequently, the authors must provide a Broader Impact Statement that explicitly addresses these risks, clarifies that the confidence score is a measure of model uncertainty and not a guarantee of factual truth, and strongly warns against deploying the system in high-stakes environments without external human review to counteract this potential erosion of critical oversight.

**Claims And Evidence:**

No

**Claims Explanation:**

The claims of efficacy are currently not fully supported due to a critical dependency on an unverified ground truth function (h). All performance measurements, including the evaluation of individual techniques and the tuning of the ensemble's weights, rely on a process, h(y_i), used to "grade" LLM responses and determine the presence of a hallucination.

The paper lacks clarity regarding the precise nature and accuracy of this function h. Furthermore, if this function is already highly effective at predicting hallucinations, the authors must clarify the necessity of developing separate UQ techniques or the complex ensemble model to approximate a solution already provided by h. This crucial methodological step needs to be fully described and justified.

**Requested Changes:**

1. Justify and Detail the Ground Truth Function (h): The authors must fully describe the ground truth function (h) used to label hallucinations. This description must include a detailed justification for its accuracy and a clear, logical explanation for why this function cannot be directly employed as the primary hallucination predictor, thus necessitating the development of the UQ techniques and ensemble model.

2. Integrate the Prompt (x) into the Classifier Input: The proposed hallucination classification or uncertainty quantification mechanism should be revised to include the input prompt (x) as an input, rather than relying solely on the LLM's output (y)

---

> ### Author Response · Authors · 2025-10-08
> **Requested changes incorporated in updated manuscript**
>
> Thank you for the thoughtful and actionable feedback. We have revised the paper to address each of your points.
>
> ### **Justifying and Detailing the Ground Truth Function h**
> Thank you for this suggestion. We have revised the paper to define, justify, and clarify the use of the ground-truth function $h$ and to explain why it cannot serve as the deployed predictor.
> 1. **Why h is not a predictor**: We added a short paragraph clarifying that while direct comparison to $y_i^∗$ yields the most accurate hallucination label, $y_i^∗$ is unavailable at generation time in deployment. Consequently, $h$ is used offline for evaluation (and, where applicable, tuning), and our goal is to approximate it at runtime via uncertainty signals (Sec. 3.1).
> 2. **Definition and role of h**: We now define explicitly the grader function $h(y_i; y_i^∗, x_i)$ as a function that grades a response against a correct reference answer. (Sec. 3.1; notation propagated in Sec. 3.5 and Appx. C).
> 3. **Task-specific graders**: We added Appx. D detailing the concrete graders of LLM responses against the QA answer keys for the tasks in our study: (a) math questions graded by integer equality, (b) multiple choice graded by letter match, and (c) short-answer graded by presence of any acceptable answer string under simple normalization.
>
> ### **Integrate the Prompt (x) into the Classifier Input**
> We revised all scorer definitions to explicitly condition on prompt $x_i$ for response $y_i$. These changes appear throughout Sec. 3 (Hallucination Detection Methodology) and Appx. C (Ensemble Tuning).
>
> ### **Broader Impact Concerns: risk of over-reliance on confidence scores**
> In **Discussion** → **Using Confidence Scores**, we added: "...a confidence score reflects model uncertainty rather than ground-truth-based correctness, and imperfect calibration can induce over-reliance in high-risk settings. We therefore caution against using confidence scores as decision guarantees in domains such as medicine, law, or finance without external human review and domain-specific safeguards."
>
> We appreciate your suggestions, which have improved clarity around $h$, made the $x$-conditioning explicit, and strengthened our broader-impact guidance.

---

### Decision · Action_Editor_3mm7 · 2025-10-31

**Recommendation:** Accept with minor revision

**Additional Comments:**

Overall the paper provides a valuable contribution to the uncertainty quantification literature and does a very nice job surveying different methods and under a standardized setup, evaluating them, and providing a tool kit for other researchers to build on.

I suggest to address the points above before publication. In addition, more minor points:

5. in the NCP metric, since NLI is an asymmetric measure, wouldn't it be more intuitive to use only the score for $\tilde{y}_{ij}$ contradicting $y_i$ since we are only interested in the correctness of $y_i$?
6. minor notation: eq.1: $\theta$ is more commonly used for input-agnostic learned parameters. Assuming $x\in\theta$ is a bit unconventional

**Audience:**

Yes

**Audience Explanation:**

The paper's extensive evaluation setup is valuable (considering the in-domain disclaimer) and the open-source repository could help others leverage the same methods for future studies. However:

4. Some of the models reported in the paper are very outdated and not even available anymore. While it's understandable that in the current LLM environment model's lifetime is relatively short, at least in the time of publication I think the paper should use recent enough models that are still available. This will ensure the relevance of the presented results and also enable other immediate followup studies to properly build on it. In addition, it will be nice to understand whether there are any plans to actively maintain the open-source repository and keep it updated.

**Claims And Evidence:**

Yes

**Claims Explanation:**

The paper studies a healthy suite of UQ methods for LLM predictions, and suggests to ensemble them with learned weights. The paper is also paired with a code repository (to be open-sourced) for reproducing the reported results (though some of the models are outdated). Overall, the main claims of the paper are well supported by a comprehensive set of experiments  over 6 populate benchmarks including numerical, multi-choice, and free text. The experiments demonstrate the ensemble approach mostly performs the best in confidence assessment in the examined experimental setup.

Therefore, overall the paper is written well and provides clear evidence. However, there are a few points that should be addressed for publication:

1. The term zero-resource is inaccurate (as pointed out by reviewer DY7K) since many of the methods require running additional models. Perhaps "closed-book" is the term that better matches the author's description of the setup.
2. The terms uncertainty quantification and hallucination detection are used interchangeably. However, they are not the exactly the same (see aleatoric vs. epistemic uncertainty). The paper should be more accurate about what it is studying.
3. The evaluation setup uses cross-validation over the test set, therefore only presenting in-domain results. In UQ specifically, OOD is a very likely practical setup. It would be good to either clarify the paper's scope, and/or to add generalization studies between the different datasets.

---

> ### Author Response · Authors · 2025-11-11
> **Requested changes reflected in camera-ready manuscript**
>
> Thank you for the close read and constructive guidance. We have addressed each point below and made corresponding edits to the manuscript.
>
> ---
>
> ### 1) “Zero-resource” terminology → “Closed-book”
>
> **Action taken:** We revised the terminology throughout to replace “zero-resource” with “closed-book,” to more accurately describe that our framework does not use retrieval, external corpora, or internet access at scoring time.
>
> ---
>
> ### 2) UQ vs. hallucination detection
>
> **Action taken:** We explicitly distinguish UQ from hallucination detection and state that we use UQ-style signals to produce response-level confidence for detection, without decomposing aleatoric/epistemic uncertainty.
>
> **Text updates:**
> - **Footnote 1 (introduction)**:
> > We employ UQ-style signals as a means to produce response-level confidence for hallucination detection, rather than pursuing uncertainty quantification as an end in itself. We do not separate aleatoric and epistemic uncertainty.
> - **Last paragraph of problem statement (Section 3.1)**:
> > We adapt techniques from the uncertainty-quantification literature to compute response-level confidence for generation-time hallucination detection. Acknowledging the distinction between uncertainty quantification and hallucination detection, we study UQ-based confidence scorers for this purpose and do not distinguish between aleatoric and epistemic uncertainty.
>
> ---
>
> ### 3) In-domain evaluation; OOD scope
>
> **Action taken:** We clarified that results are **in-domain**, and added an explicit limitation noting that we do not evaluate OOD generalization or cross-dataset transfer of learned weights.
>
> **Text updates:**
> - **Section 3.5 (Ensemble scorer)**
> > Because scorer performance depends on both the dataset and the underlying LLM, the ensemble weights are tuned per use case (chosen LLM and dataset), and the resulting weights are intended for in-domain deployment.
> - **Section 4.2 (Experiment results, footnote)**
> > Scorer performance depends on both the dataset and the LLM, so we tune the ensemble for a specific LLM–dataset pair. Our experiments reflect this in-domain setup; we do not evaluate out-of-distribution generalization across datasets or domains.
> - **Section 5 (Discussion -> Limitations and Future Work)**
> > ...while the ensemble approach is designed to optimize in-domain performance, we have not evaluated out-of-distribution generalization, including cross-dataset transfer of learned weights; we leave this to future work.
>
> ---
>
> ### 4) Model recency and repository maintenance
>
> **Model Recency:** We have updated our experiments so that all LLMs used are current. This includes the following: GPT-4o, GPT-4o-mini, Gemini-2.5-Flash, Gemini-2.5-Flash-Lite. Overall, the results are consistent with the previous version:
>  - Ensemble outperforms individual scorers (20/24 AUROC; 17/24 F1-score)
>  - NLI is a high-performing black-box method (best black-box performance in 13/24 AUROC, 18/24 F1-Score)
>  - LLM-as-a-Judge performance is positively related to the LLM's accuracy on those same datasets (evident from Gemini-2.5-Flash on math benchmarks and GPT-4o on short-answer benchmarks)
>  - Gains from additional sampled responses are positive but diminish as more responses are sampled.
>  - The two white-box scorers pefform approximately equally.
>
> **Repository maintenance:** We added an explicit maintenance note in the Conclusion section to signal ongoing support for the codebase.
> > The `uqlm` library implements all scorers presented and evaluated in this work. The repository is actively maintained. We welcome issues and pull requests and will continue to update integrations, add new scorers as research advances, and refresh examples as model and provider APIs change.
>
> ---
>
> ### 5) NCP metric asymmetry (entailment/contradiction direction)
>
> **Action taken:** We added a short note discussing the asymmetry and why we report the symmetric form while acknowledging that the directional variant is a reasonable alternative.
>
> **Text updates:**
> - **Section 3.2 (Black-Box UQ Scorers), footnote**
> > We note that NLI is an asymmetric measure. Manakul et al. (2023) propose a one-directional variant of NCP. We follow Chen & Mueller (2023) in using the bidirectional formulation to allow for more flexibility in detecting contradictions.
>
> ---
>
> ### 6) Notation: $\theta$ used for input-agnostic parameters
>
> **Action taken:** We updated the notation to so that conditioning context variables (prompts, sampled responses) are no longer denoted by $\theta$. (Section 3.1)
>
> ---
>
> We appreciate the careful feedback and believe these changes resolve the concerns. Please let us know if you have any further feedback.

---

> > ### Comment · Action_Editor_3mm7 · 2025-11-12
> >
> > Thank you very much for your detailed revision and addressing all comments. Can you please just update the openreview abstract to match the pdf?

---

> > > ### Author Response · Authors · 2025-11-12
> > > **Openreview abstract updated**
> > >
> > > We have updated the openreview abstract to match the PDF. Thank you again.